# Cellulose from Annual Plants and Its Use for the Production of the Films Hydrophobized with Tetrafluoroethylene Telomers

**DOI:** 10.3390/molecules27186002

**Published:** 2022-09-15

**Authors:** Sergey A. Baskakov, Yulia V. Baskakova, Eugene N. Kabachkov, Galina A. Kichigina, Pavel P. Kushch, Dmitriy P. Kiryukhin, Svetlana S. Krasnikova, Elmira R. Badamshina, Sergey G. Vasil’ev, Timofey A. Soldatenkov, Victor N. Vasilets, Filipp O. Milovich, Alexandre Michtchenko, Oksana V. Veselova, Vasiliy A. Yakimov, Svetlana N. Ivanova, Yury M. Shulga

**Affiliations:** 1Institute of Problems of Chemical Physics, Russian Academy of Sciences, Chernogolovka 142432, Russia; 2Chernogolovka Scientific Center, Russian Academy of Sciences, Chernogolovka 142432, Russia; 3Faculty of Fundamental Physical and Chemical Engineering Lomonosov, Moscow State University, Moscow 119991, Russia; 4Branch of N.N. Semenov Federal Research Center of Chemical Physics, Russian Academy of Sciences, Chernogolovka 142432, Russia; 5National University of Science and Technology MISIS, Leninsky pr. 4, Moscow 119049, Russia; 6Instituto Politécnico Nacional, SEPI-ESIME-Zacatenco, Av. IPN S/N, Ed. 5, 3-r piso, Ciudad de México 07738, Mexico; 7Resource Innovation Center “Center for Biodesign and Bio-Nanomaterials”, The Federal State Budgetary Educational Institution of Higher Education “Russian State University Named after A.N. Kosygin (Technology. Design. Art)”, Malaya Kaluzhskaya St. 1, Moscow 119071, Russia

**Keywords:** cellulose separation, hogweed, telomers of tetrafluoroethylene, hydrophobization

## Abstract

Cellulose **HogC** was produced by the modified traditional method with 35% yield from the stem of Sosnovsky hogweed and was characterized by elemental analysis, infrared (IR) spectroscopy, powder X-ray diffractometry, differential scanning calorimetry (DSC) and X-ray photoelectron spectroscopy (XPS). For **HogC**, the degree of crystallinity (approximately 70%) and the glass transition temperature (105–108 °C) were determined. It was found that the whiteness characteristic in the case of **HogC** was 92% and this significate was obtained without a bleaching procedure using chlorine-containing reagents. In this paper, the possibility of hydrophobization of **HogC** films by treatment with radiation-synthesized telomers of tetrafluoroethylene is shown. It was found that the contact angle of the telomer-treated cellulose film surface depended on the properties of the telomers (the chemical nature of the solvent, and the initial concentration of tetrafluoroethylene) and could reach 140 degrees.

## 1. Introduction

Usually, cellulose is obtained from pure wood and waste paper. However, the main source of cellulose fibers at present should be considered to be wood pulp. Cellulose fibers in wood are interconnected by a rigid three-dimensional polymer, lignin, which occupies up to 30% of the wood mass. Methods for isolating cellulose from wood pulp are well known [1,2,3,4,5,6]. However, what if there is little or no source material (wood)? It is believed that instead of trees, which must grow for several years before becoming a source of cellulose, annual plants can be used [7,8,9,10]. Within this approach, the search for sources of cellulose should be directed to plants with a large biomass, the production of which can be quite cheap. For obvious reasons (high cost of production), plants such as cotton should be excluded from this search, even though cotton is 100% cellulose.

Sosnovsky hogweed is a large herb. In the area of the Caucasus foothills, the plant reaches approximately 1–1.5 m in height, whereas in Poland its size is significantly larger, up to 3–3.5 m. It has a thickly ridged, hollow stem, up to 12 cm in diameter. The leaves are palmate and reach up to 2 m in length [11]. The biomass can reach 2.3–4.6 kg/m^2^ [12]. Even with a minimum yield of 2 kg/m^2^, we get 20 tons of biomass from one ha of hogweed. The annual growth of wood in our latitudes is about 4 m^3^/ha, which gives about 4 tons of biomass (this is an average value). Therefore, the production of pulp from hogweed (even at 35% yield, as in our case, see below) appears to be more efficient in terms of land use compared to production from wood.

Many natural materials are composites. For example, wood fibers are composed of cellulose microfibrils embedded in an amorphous matrix of lignin and hemicellulose. It is also known from the literature that hogweed, in addition to photosensitive substances from the furanocoumarin group, which, when illuminated, can cause serious skin damage, also contains substances, due to which it was previously classified as a silage plant [13]. Hogweed contains lignin macromolecules, which have a “high antioxidant activity, comparable to the activity of recognized antioxidants” [14]. It is observed that industrial ethanol and cellulose can also be isolated from hogweed.

This work is devoted to the description of our experience in obtaining and studying the properties of cellulose isolated from dried hogweed stems (**HogC** is an abbreviation for hogweed cellulose). The current study can be considered relevant from the point of view of studying the prospects for using the biomass of this widespread harmful plant. Literary searches have shown that the methods for isolating cellulose from hogweed have not been well studied yet.

Cellulose is one of the most prevalent organic polymers with inexhaustible natural resources. However, its hydrophilic nature makes cellulose sensitive to water, which limits its durability [15]. Therefore, hydrophobic modification of the surface of cellulose products is of great importance. Earlier, fluoropolymers and telomers of tetrafluoroethylene (TFE) were used to modify cellulose-containing materials [16]. In this work, the surface of the HogC film was treated with solutions of TFE telomers having the general formula R_1_(CF_2_CF_2_)_n_R_2_ (R_1_ and R_2_ are fragments of solvent molecules, n is the chain length). The telomeres used were obtained by the radiation-initiated telomerization of TFE in carbogal [17], in binary solvent Freon 113+ ammonia [18], and acetone [19]. Due to fact they possessing different chain lengths and end groups, it was possible to obtain a protective hydrophobic coating.

This work describes a simple procedure for obtaining the cellulose film and presents the results of studying the surface of hydrophobized **HogC** film by the XPS method and the results of measuring the contact angle for a water drop on the film surface.

## 2. Experimental

Reagents: Nitric acid (concentration 70% GOST 11125-84) was obtained from JSC Base No. 1 of Chemical Reactives, Staraya Kupavna, Russia. Hydrogen peroxide (concentration 33%, TU 20.13.63-207-44493179-2016) and potassium hydroxide (chemically pure) were provided by EEEKOS-1 LLC, Russia, Staraya Kupavna. Tetrafluoroethylene (C_2_F_4_, TFE), Flutec PP3 (perfluoro-1,3-dimethylcyclohexane, C_8_F_16_) and trifluorotrichloroethane (C_2_F_3_Cl_3_, Freon 113), produced by the Polymer Plant of the Kirovo-Chepetsk Chemical Integrated Works, acetone (C_3_H_6_O) (Aldrich, Darmstadt, Germany) and gaseous ammonia NH_3_ (United Chemical Company Shchekinoazot, Tula, Russia) were used for the radiation synthesis of telomers. The reagents were used without further purification. Double-distilled water was used in the work. 

The stems of Sosnovsky hogweed were collected in the area of Pushkino, Moscow region. The collected stems were dried to constant weight at 60 °C in a drying oven. After that, the stems were crushed into fragments 2 ÷ 5 cm^2^ in size. An amount of 30 g of chopped hogweed was washed 3 times with water. Then, it was poured into a two-necked flask with a volume of 2 L. The amount of 1.5 L of a 3% nitric acid solution was poured into the flask. The flask was thermostated to 80 °C and stirred until the stem fragments disintegrated into separate bundles of fibers (approximately 24 h). The resulting yellow-brown fibrous material was collected by filtration and washed to remove free nitric acid. The resulting mass was loaded into a flask and 1.5 L of a 2% KOH solution was added, the mixture was kept under stirring at a temperature of 60 °C for another 2 h. After filtration and washing, light gray cellulose was obtained. The resulting cellulose was additionally bleached in 1.0 L of a 3% hydrogen peroxide solution with the addition of 10 g of KOH. The reaction was carried out at 50 °C for 30 min. Then, the cellulose was washed and dried on a freeze-dryer (model FDS5512 from IlShin BioBase, Dongducheon, Korea) at a pressure of 5 mTorr and a temperature of −55 °C (the cellulose isolation scheme is also shown in Figure 1). 

Film Preparation Technique. An aqueous suspension of cellulose **HogC** (concentration 20 g/L) was mixed well. The resulting mixture was poured into a 200 × 100 mm glass mold and leveled off to prevent perimeter film thickness fluctuations. After three days, the dried film was separated from the mold.

To prepare solutions of tetrafluoroethylene telomers, the following chemicals were used: gaseous tetrafluoroethylene C_2_F_4_ (TFE) containing 0.02% impurities, acetone, trichlorotrifluoroethane C_2_F_3_Cl_3_ (Freon 113), gaseous ammonia NH_3_ and perfluoro-1,3-dimethylcyclohexane (Flutec PP3, C_8_F_16_). All the chemicals were used without additional purification. Radiation-initiated polymerization was carried out in sealed glass ampoules. The samples were prepared according to the standard procedure: a certain amount of solvent was placed in a 10-mL glass ampoule, dissolved air was removed, the required amount of tetrafluoroethylene was condensed into the ampoule at 77 K, and the ampoule was sealed. When the reaction was carried out in a binary solvent, Freon 113–NH3, gaseous ammonia was frozen in an ampoule at 77 K. The system was stirred at room temperature and irradiated with ^60^Co γ-rays at the Gammatok-100 unique research facility at a dose rate of 3.2 Gy/s. The TFE concentration in acetone was 1.0 ± 0.01 mol/L, that in Freon 113 was 0.5 ± 0.01 mol/L, in Flutec PP3 0.06–0.3 ± 0.01 mol/L and the ammonia concentration was 0.11 ± 0.01 mol/L. The concentrations of the telomer solutions obtained were determined gravimetrically after removing the solvent from the reaction mixture. The measurement accuracy was ± 0.5% wt %.

The solutions of telomers were applied to samples by impregnation. The treatment of the samples included the following operations: sample immersion in a telomer solution (30–40 s), and drying at 70 °C (40 min). The telomer amount applied to a sample was controlled by gravimetry. The concentration of impregnating solutions in acetone and Freon 113 was ~3.0–4.0 wt %, and in Flutec PP3 ~0.35, 1.1, 1.8 wt %.

Elemental analysis was carried out on a CHNS analyzer “Vario Micro cube” (Elementar GmbH, Hanau, Germany). The IR spectra (resolution 1 cm^−1^, number of scans 32) were recorded at room temperature in the range of 400–4000 cm^−1^ on a Perkin-Elmer “Spectrum Two” Fourier-transform IR spectrometer (Waltham, Massachusetts, United States) with an ATR attachment. The water contact angle was measured on an OCA 20 instrument (Data Physics Instruments GmbH, Fielderstadt, Germany) at room temperature. X-ray Photoelectron spectra (XPS) were obtained using a Specs PHOIBOS 150 MCD electron spectrometer (SPECS GmbH, Berlin, Germany) and X-ray tube with an Mg cathode (hν = 1253.6 eV). The vacuum in the spectrometer chamber did not exceed 4 × 10^−8^ Pa. The spectra were recorded in the constant transmission energy mode. XPS background subtraction was carried out, according to the Shirley method, and the XPS spectra decomposition was performed, according to the set of mixed Gaussian/Lorentz peaks in the framework of the Casa XPS 2.3.23 software (Casa Software Ltd., Teignmouth, UK). The quantification of atomic content was performed using sensitivity factors provided by the elemental library of CasaXPS. Differential scanning calorimetry curves were recorded using a Mettler-Toledo DSC 822 instrument (Mettler-Toledo, Zurich, Switzerland). Samples weighing 7–10 mg were placed in an aluminum ampoule, which was in the process of measurement in an argon atmosphere at a flow of 50 mL/min. Heat release was measured in the temperature range from −10 to +200 °C at a minimum heating rate of 5 °C per min. The glass transition temperature (Tg), with an experimental error ±5 °C, was calculated using Mettler-Toledo software. Diffraction studies were carried out on a D2 PHASER X-ray Powder Diffractometer from Bruker (Germany). All NMR experiments were performed on a Bruker Avance-III-400 (Bruker, Ettlingen, Germany) operating at 9.4 T (400.2 MHz for ^1^H and 100.6 MHz for ^13^C) and equipped with a 3.2 mm solid-state magic-angle-spinning (MAS) probe head. Measurements were conducted at 297 K with a MAS spinning rate of 15 kHz. ^1^H-^13^C Cross-Polarization Magic-Angle Spinning (CP/MAS) spectra were acquired using a standard pulse sequence with 70–100% ramped contact pulse on 1H. The length of ^1^H 90° pulse was 2.7 us, the same rf amplitude was used during the contact and SPINAL-64 decoupling. A contact time of 1.2 m, acquisition time of 35 m and recycle delay of 5 s and 13,500 scans were used. The observed intensities in the CP/MAS spectra might be distorted, due to different relaxation characteristics of the regions with different states of order and mobility. 

## 3. Results

### 3.1. Elemental (C, H, N, S) Analysis

The chemical formula of cellulose is quite simple, C_6_H_10_O_5_. The results of the elemental analysis (Table 1) are somewhat different from this formula. In the investigated cellulose sample, the O/C ratio was higher than if it followed the chemical formula. The composition of the sample under study may have contained impurities that were not determined during the performed analysis. Indeed, in the composition of **HogC**, the presence of sulfur can be noted, which passed into the test sample from the original hogweed powder (Table 1). As you know, sulfur is found in plants in two main forms: oxidized in the form of inorganic sulfate and reduced (amino acids, proteins). The described isolation procedure was not effective enough to remove sulfur-containing compounds. The presence of other impurities was also possible, which could also have increased the estimate of the oxygen content.

### 3.2. Technical Characteristics of HogC

To complete the picture of the technical characteristics of the obtained cellulose, it was tested, according to standard methods recommended in the laboratories of factories for the production of viscose. Before testing, loosening was carried out, during which a noticeable amount of small, dusty fibers poured out. These wastes could reach 15 wt %. We noted here that the so-called dust fibers were due to the cellulose of the parenchyma. In principle, the authors know the technique for separating parenchyma and long fibers, which they will present in another publication. Below are the results of these tests (Table 2). First of all, such a characteristic as whiteness attracts attention. A high value of this index was obtained without a bleaching procedure using chlorine-containing reagents.

### 3.3. IR Spectra 

The IR spectra of **HogC** and the powder obtained by grinding dry hogweed stems are shown in Figure 2. It can be seen that both spectra have a broad absorption band (AB) in the range 3690–3050 cm^−1^, which is caused by stretching vibrations of O-H bonds. The large half-width of this AB indicated strong hydrogen bonds in the samples under study. In both spectra, there was an AB due to stretching vibrations of C-H bonds (range 2988–2812 cm^−1^). There were significant differences between the compared spectra. Thus, in spectrum 1, one could see the absorption band at 1736 cm^−1^ (stretching vibrations of C=O bonds), which was absent in the spectrum of **HogC**.

Let us now compare the spectrum of the cellulose obtained by us with the literary spectra of cellulose in the field of “fingerprints”. Thus, in the widely cited work, [20], IR spectra were shown for three crystalline and one amorphous cellulose samples. Table 3 compares the position for four ABs of these cotton samples (Cr-I, Cr-II, Cr-III, and Am) [20] with those for the cellulose sample obtained by us (**HogC**). It can be seen that the positions of two ABs in the **HogC** spectrum (1160 and 1105 cm^−1^) differed from the positions of these bands in the crystalline and amorphous samples.

Typically, most cellulosic materials are composed of crystalline and amorphous domains. The interaction of solid cellulosic materials with water and other reagents first occurs in non-crystalline domains and on the surface of cellulose crystallites. It was found that the amorphization of cellulose led to a significant simplification of the IR spectrum due to a decrease in the intensity, or even disappearance of bands characteristic of crystalline domains. Nowadays, IR spectroscopy is often used to assess the degree of crystallinity of cellulose (see, for example, [21]). Thus, the absorption band at 898–983 cm^−1^ was usually positioned as an “amorphous” absorption band, an increase in its intensity occurred in the amorphous samples. As an alternative to the amorphous band, an absorption band at 1430–1420 cm^−1^ was used. The ratio of the intensities of these bands (marked with asterisks in Figure 2) was an indicator of the crystallinity of the studied cellulose samples. In the case of **HogC**, the A_1430_/A_893_ ratio was approximately 0.91. For cotton cellulose, this ratio was 1.86. Hence, the degree of crystallinity, defined as A_1430_/(A_893_ + A_1430_), was 0.48 and 0.65 (48 or 65%) for **HogC** and cotton cellulose, respectively.

### 3.4. XRD

Another widely used method for evaluating the degree of crystallinity of cellulose is the X-ray diffraction method [22,23,24,25,26,27,28]. It is important to note that X-ray diffraction provides more detailed data on the features of the crystalline fraction of cellulose. The X-ray diffraction pattern of **HogC** is shown in Figure 3. The assignment of the peaks was done following the work in [22]. The region of coherent scattering, calculated using the Scherrer formula from the half-width of the most intense peak (200), was equal to 4 nm. The apparent degree of crystallinity C (in %) was determined by the Segal method [28], according to the formula:C = 100 (I_200_ − I_non-cr_)/I_200_(1)
where I_200_ is the intensity of the (200) peak, I_non-cr_ is the diffraction intensity of the non-crystalline material, which is defined as the intensity in the trough between the peaks (1$\bar{1}$0) and (200) (2θ ≈ 18 degrees). For the sample under study, the apparent degree of crystallinity turned out to be 70%, which was higher than that determined from the IR spectrum (48%).

### 3.5. Glass Phase Transition Temperature (DSC Measurement)

An essential characteristic of cellulose is the glass transition temperature (the transition of a polymer from a solid glassy state to a rubber one). Currently, the glass transition temperature (Tg) is usually determined by the DSC method [29,30,31]. Determining the exact value of Tg causes certain difficulties, since the process can occur in a wide temperature range (up to several tens of degrees). The Tg value is defined as the midpoint of this temperature range. Figure 4 shows DSC curves for one of the studied **HogC** samples in the region of the proposed transition. It can be seen that during the first heating run, the transition region overlapped with an intense broad endothermic peak, which was in accordance with literature data [29,30,31], and was associated with the evaporation of free water, which is always present in polymers containing hydrophilic groups. It is generally believed that the amorphous part of the cellulose almost entirely determines the sorption of water. Therefore, the area of the endothermic peak could be used to assess the relative degree of cellulose amorphization in a series of samples that were previously incubated for a long time at the same humidity.

The glass phase transition opened during subsequent heating-cooling cycles. The Tg values determined in the second and third heating runs were 105 and 108 °C, respectively. Literature data for Tg in the case of cellulose were in the range from 220 °C [32] to −137 °C [33]. The specific value depends in a complex way on the parameters, such as the degree of crystallinity and the water content (C_H2O_). By weighing the sample before and after heating to 120 °C, it was possible to determine the content of free water evaporating during the first pass, which was 1.7 wt %. Note that in addition to free water, which can be removed by heating to 100–120 °C, bound water may be present in the cellulose. For the studied sample, HogC, the degree of crystallinity from XRD was close to 70%. In accordance with the dependence Tg = Tg (C_H2O_) presented in [29], it was possible to estimate the concentration of bound water in our sample. It was 3–4 weight %.

### 3.6. Solid State NMR Spectra

To study the structure of **HogC** cellulose, its solid-state CP/MAS ^13^C NMR spectrum was also obtained (Figure 5). The assignment of peaks was carried out, according to data in the literature [34,35,36,37]. The most intense split peak, with the maxima at 77.1 and 72.6 ppm, were due to carbon atoms C2, C3 and C5. The assignment of the remaining peaks is indicated in Figure 5. The spectra were approximated by the mixed Gauss/Lorentz functions. The results are shown in Figure 5, as well as in Table 4. Comparison of the signal intensities corresponding to various atoms in the structure showed good agreement with the structure of the compound. The intensity ratio of the signals C1:C4:(C2 + C3 + C5):C6 matched with good accuracy to the intensity ratio 1:1:3:1 corresponding to the last column in Table 4. As a rule, the peaks corresponding to C1 and C4 atoms were used to determine the crystallinity of cellulose [38,39,40]. The peaks corresponding to C1 atoms were not resolved and asymmetry of the total line was observed only as a shoulder in the region of smaller chemical shifts. For us, a more informative region in the ^13^C CP/MAS NMR spectra turned out to be the region of peaks due to the C4 atom, with the maxima at 89.1 and 84.4 ppm. According to [38,39,40], the region 92–86 ppm corresponds to crystalline cellulose, and the region 86–80 ppm corresponds to amorphous cellulose. Spectral fit with Gauss/Lorentz lines in this region showed that the ratio of the peak area in the range of 92–86 ppm (peaks 3 and 4 in Table 4) to the total peak’s area of C4 signals (peaks 3, 4, 5, 6 in Table 4) was equal to 38.4% for the sample under study. This was less than the apparent degree of crystallinity calculated from the IR and XRD data.

### 3.7. Contact Water Angles (CWA)

The **HogC** film is hydrophilic, so a drop of water wets it completely. Treatment of the surface of the **HogC** film with solutions of TFE telomers, the synthesis of which is described in detail in [17,18,19], made it hydrophobic. Contact water angles (CWA) for 5 samples of films treated with telomers, which were obtained in different solvents and with different TFE concentrations, are presented in Table 5.

It can be seen from the table that all samples were hydrophobic, and the CWA values depended both on the TFE concentration at which the telomer solutions in carbogal were obtained and on the solvent used as telogen in the synthesis. This was because the chain length of the resulting telomer depended on the initial TFE concentration and the chemical nature of the solvent, on which, in turn, the quality of the obtained hydrophobic coating depended. It was shown earlier that it is advisable to use TFE telomeres with a chain length of about 15–50 TFE units to obtain coatings with high hydrophobicity [41]. Telomers with a chain length of fewer than 10 units are not suitable for creating hydrophobic coatings. Probably, the small contact angle obtained for sample 1 was associated with the fact that telomeres with a relatively short chain length were formed during the synthesis of telomers in carbogal with an initial TFE concentration of 0.06 mol/L. In addition, a small amount of telomer was applied at a low concentration of the used solution (0.35 wt %), and it was necessary to increase the amount of impregnation. Usually, solutions with a concentration of 3.0–4.0 wt % were used for impregnation. 

### 3.8. X-ray Photoelectron Spectra (XPS)

A sample with the maximum CWA value (sample No. 2, Table 5) was selected to study the surface composition by the XPS method. The elemental composition of the near-surface layer of the cellulose film before and after telomer treatment is presented in Table 6. On the surface of the untreated film the O/C ratio was less than that of pure cellulose. In our opinion, this occurred due to the sorption of hydrocarbon contaminants on the film surface. The presence of silicon in a sufficiently high concentration should also be noted. In principle, a certain amount of silicon is present in the ash of trees and plants. For example, the SiO_2_ content in the ash of rice straw can reach 85 mass % [42]. We did not find such data for hogweed in the literature. Therefore, we obtained a survey spectrum of the ash of dried hogweed stems and calculated the corresponding elemental composition (Figure 6, Table 7). The data presented that silicon is a part of the hogweed. However, its concentration in ash is less than on the surface of **HogC** film. It could be argued that the source of silicon contamination of the cellulose film surface was the glass mold in which the film was formed and dried. It was observed that the **HogC** film was more difficult to separate from glass than the film of other cellulose-based materials (e.g., ashless filter). The XPS spectrum was recorded from the smoother side of the film, which was in contact with the glass substrate.

The high-resolution spectra for the C1s line of the **HogC** film before and after telomer treatment can be seen in Figure 7. In accordance with the literature data (see, for example, [43,44,45,46], the C1s line of cellulose was described by four peaks, which were caused by carbon atoms that did not have bonds with oxygen: (C1), which had one bond with an oxygen atom, (C2), which had two bonds with the oxygen atom/atoms, (C3), which had three bonds with the oxygen atoms (C4), respectively. The relative positions of the peaks are somewhat different for different authors. We present here the rounded values of the binding energies for C1, C2, C3, and C4 atoms, taken from [39]: 285, 287, 288, and 289–290 eV, respectively. Note here that, as a rule, the intensity of the C4 peak is low and, therefore, is not given by some authors. Note also that in ideal cellulose, there are no carbon atoms, only carbon or hydrogen atoms in the nearest environment (see the chemical structure of the polysaccharide and the assignment of peaks in Table 7). However, in the XPS spectra of real cellulose samples, there is always a peak, due to carbon atoms of this type. Often, the assumption of a layer of hydrocarbon contaminants on the cellulose surface is insufficient to explain the origin of the peak under discussion. There may be other reasons; for example, insufficient degree of purification from lignin, where such carbon atoms are present.

After treatment with a telomere solution, the most intense peak in the spectrum of the film was the peak due to carbon atoms of the CF_2_ groups (Figure 7, Table 8). We also noted here that the decomposition gave rise to the C6 peak, which could be attributed to end groups. The C5/C6 ratio can estimate the average number of tetrafluoroethylene molecules that participated in telomer formation [43]. Finally, from the data presented in Table 8, it followed that the layer analyzed by the XPS method consisted of almost 80% telomer molecules, which explained the reason for the hydrophobicity of the surface of the **HogC** film after this treatment.

## 4. Conclusions

In summary, cellulose **HogC** was produced by an attractive method from the dry stems of Sosnovsky hogweed employing successive grinding, acidizing (3% HNO_3_, 24 h, 80–90 °C), alkaline treatment (2% KOH, 2 h, 60–70 °C), bleaching (3% H_2_O_2_ + 1% KOH, 20 min, 60–70 °C) and freeze-drying. In this sense, our work stands in line with other modern works [47,48,49,50,51,52,53,54,55], which propose the use of plants, rather than trees, for the production of cellulose. The composition, crystalline structure, vibrational response, thermal stability and some other technical indices of the cellulose was studied using modern techniques of material characterization. It was found that such a characteristic as whiteness in the case of **HogC** was 92% and this significate was obtained without a bleaching procedure using chlorine-containing reagents. When bleaching with hydrogen peroxide, the direct bleaching effect on cellulose, or rather, on its residual lignin, is exerted by the peroxide ion resulting from the hydrolytic dissociation of peroxide: H_2_O_2_ ↔ H^+^ + HO_2_^−^. An alkaline environment, since hydrogen ions are neutralized in this case, and the equilibrium of the hydrolysis reaction shifts to the right to facilitate the dissociation of peroxide. Therefore, bleaching with peroxide was carried out in an alkaline environment, at pH = 10–11. The use of hydrogen peroxide as a bleaching agent made it possible to obtain bleached pulp from Sosnowski’s hogweed with a high level of whiteness. At the same time, as an advantage over the classical scheme of chlorine bleaching of pulp, in addition to higher environmental friendliness, the method is simple, because the process is performed in just one-step. The degree of crystallinity (70% from XRD data) and the glass transition temperature (105–108 °C) were determined for the cellulose from hogweed. Further, films were made from **HogC**, and the possibility of their hydrophobization, by treatment with radiation-synthesized telomers of tetrafluoroethylene, was shown. It was found that the contact water angle of the telomer-treated cellulose film surface depended both on the chemical nature of the solvent and on the initial concentration of tetrafluoroethylene during telomer synthesis and could reach 140 degrees.

## Figures and Tables

**Figure 1 molecules-27-06002-f001:**
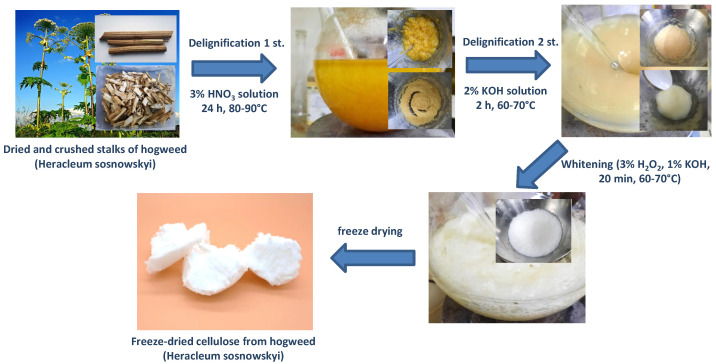
Scheme of obtaining cellulose from the stalk of hogweed.

**Figure 2 molecules-27-06002-f002:**
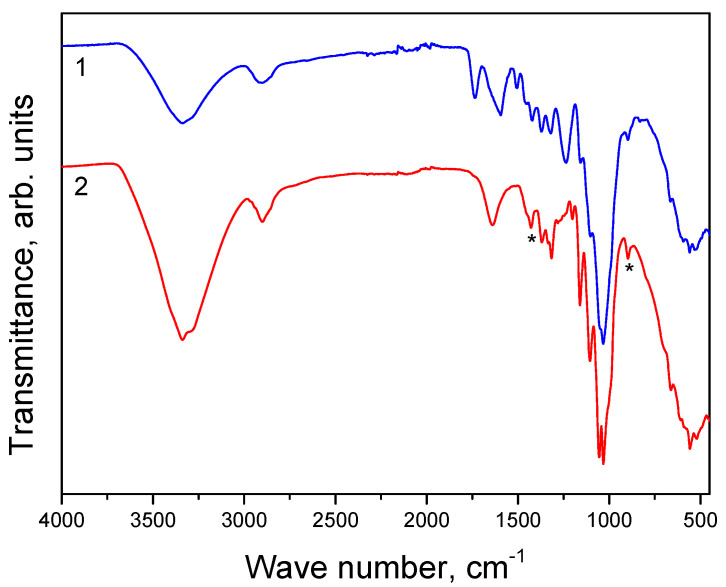
IR spectra of the powder were obtained by grinding dry hogweed stems (1) and **HogC** (2). The * sign marks the absorption bands at 1430 and 893 cm^−1^ (see below).

**Figure 3 molecules-27-06002-f003:**
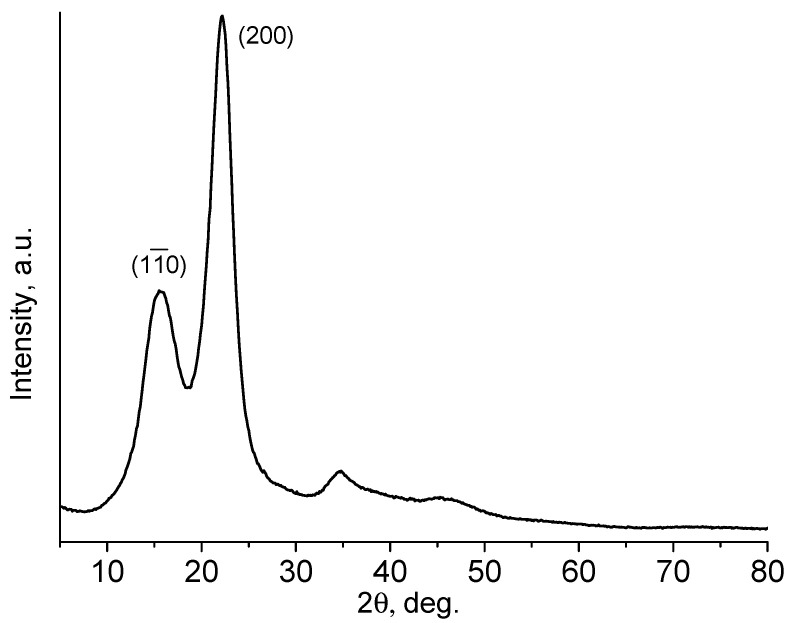
XRD patterns of **HogC** cellulose at ambient temperature.

**Figure 4 molecules-27-06002-f004:**
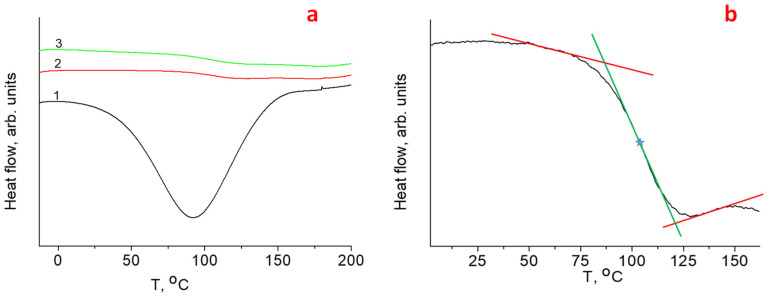
(**a**)—DSC curves for **HogC** cellulose (curve 1—the first heating of the **HogC** sample from −10 to +200 °C; curve 2—the second heating of the **HogC** sample from −10 to +200 °C; curve 3—the third heating of the **HogC** sample from −10 to +200 °C); (**b**)—illustration of the determination process for Tg value.

**Figure 5 molecules-27-06002-f005:**
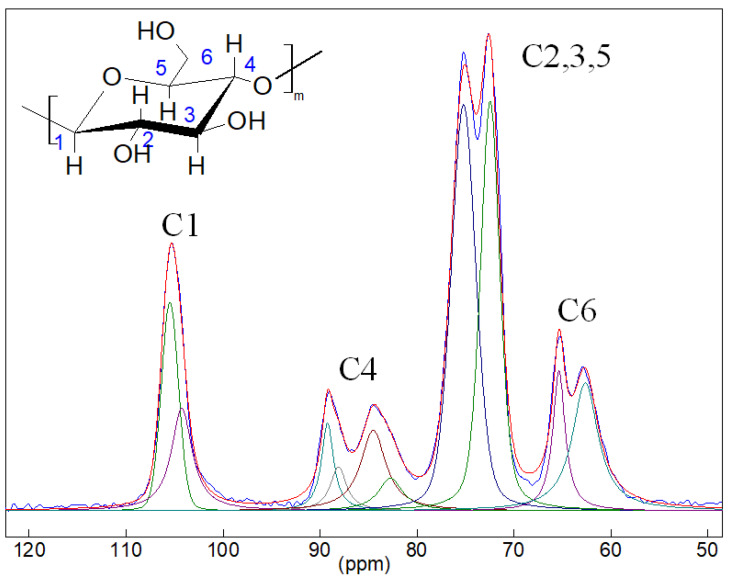
CP/MAS ^13^C NMR spectrum of the **HogC** sample. The results of the description of the spectrum by 10 mixed Gauss/Lorentz functions are presented in Table 2. The inset shows the chemical structure of cellulose. Numbers indicate carbon atoms.

**Figure 6 molecules-27-06002-f006:**
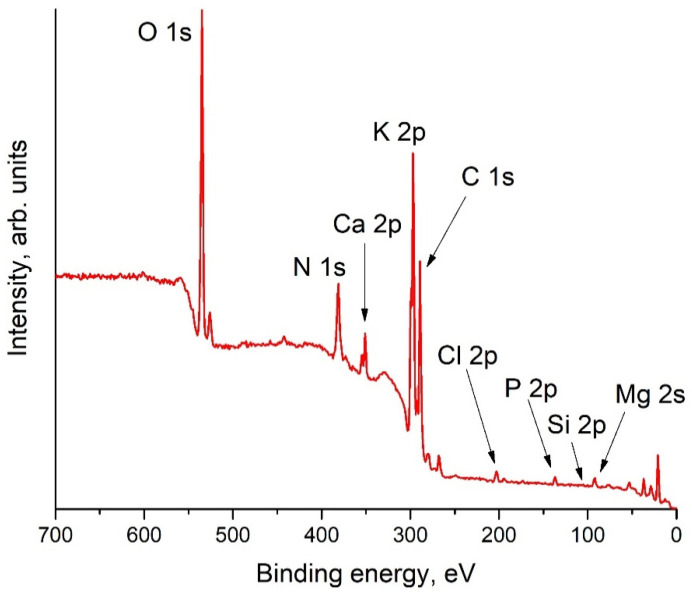
Survey XPS spectrum of ash obtained by burning hogweed in air.

**Figure 7 molecules-27-06002-f007:**
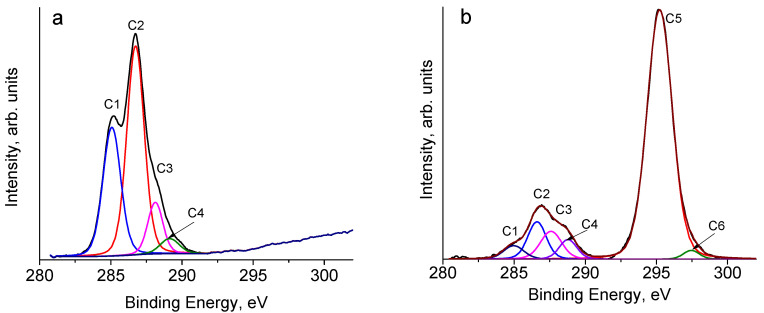
XPS spectra of HogC film before (**a**) and after (**b**) telomer treatment.

**Table 1 molecules-27-06002-t001:** Elemental composition of dry hogweed powder, **HogC** and pure cellulose (in wt %).

Sample	C	H	N	S	O ^1^
Dry hogweed powder	39.95	5.440	0.21	0.453	53.951
**HogC**	40.80	6.185	0.05	0.254	52.711
Pure cellulose ^2^	44.44	6.17	-	-	49.38

^1^—oxygen content was estimated by the formula [O] = 100−$\underset{i}{\sum }[C_{i}]$, where [*C_i_*] is the content of the *i*-th element. ^2^—calculated according to the formula C_6_H_10_O_5_.

**Table 2 molecules-27-06002-t002:** Technical specification of fluff pulp sample of **HogC**.

Index, Units	Standard Indicator	Test Results
Conditional humidity, %	8.0	5.5
Mass fraction of resins and fats, %	no more than 0.16	0.14
Whiteness, %	no less than 86.0	92.5
Absorbent capacity, g/g	no less than 15	36.2
Absorption time, s	no more than 3.5	1.8

**Table 3 molecules-27-06002-t003:** Frequencies of some absorption bands (in cm^−1^) in the IR spectra of native cotton (Cr-I), Fortisan (Cr-II), and cotton cellulose III (Cr-III), vibratory ball-milled cotton (Am), and cellulose, isolated by us from the crushed stalks of hogweed (**HogC**). Samples Cr-I, Cr-II, Cr-III, and Am were obtained and characterized in [20].

	Band	1429	1163	1111	897
Sample					
Cotton **Cr-I**		1429	1163	1111	897
Cotton **Cr-II**		1420	1156	-	893
Cotton **Cr-III**		1425	1163	1102	897
Cotton **Am**		1420	1156	-	897
**HogC**		1429	1160	1105	897
*		the CH_2_ scissoring motion	the antisymmetrical bridge C-0-C stretching	the *ν**_ai_* mode of ring stretching	characteristic of β-linked glucose polymers

* The bottom line of the Table shows the AB assignments taken from [20].

**Table 4 molecules-27-06002-t004:** Position, width, and intensity of individual peaks in the CP/MAS 13C NMR spectrum of the **HogC** sample.

Peak	Assignment (Figure 5)	Position, ppm	Width, Hz	Intensity, %	I(CJ)/I(C1)
1	C1	105.49	210	8.92	1.00
2	C1	104.26	260	7.56	
3	C4	89.23	151	3.76	0.98
4	C4	88.11	202	2.53	
5	C4	84.52	319	7.27	
6	C4	82.74	305	2.84	
7	C2, C3, C5	75.21	302	26.99	3.00
8	C2, C3, C5	72.46	236	22.22	
9	C6	65.37	154	6.13	1.09
10	C6	62.65	324	11.76	

**Table 5 molecules-27-06002-t005:** Contact water angles for **HogC** cellulose films after hydrophobization.

Sample	Solvent	TFE Concentration, mol/L	CWA, Degrees
1	carbogal	0.06	105.1
2	carbogal	0.20	143.4
3	carbogal	0.33	133.4
4	acetone	1.0	137.3
5	Freon 113 + NH_3_	0.5	132.5

**Table 6 molecules-27-06002-t006:** Surface composition (in at %) of the film (before and after telomer treatment) determined from XPS.

C-Hog Film	C	O	N	Si	F
Before treatment	62.64	34.63	0.38	2.35	-
After treatment	35.03	3.75	-	1.48	59.73

**Table 7 molecules-27-06002-t007:** Composition of the hogweed ash (in at %) determined from XPS.

Element	O	C	N	Ca	K	Cl	P	Mg	Na	Si
Conc. at. %	29.23	40.08	0.26	2.8	21.02	1.35	1.6	2.79	0.4	0.47
St.Dev.	0.266	0.294	0.132	0.105	0.168	0.075	0.144	0.259	0.2	0.142

**Table 8 molecules-27-06002-t008:** Relative peak areas (in %) from deconvoluted C 1s spectra.

C-Hog Film	C1 (C-C/C-H)	C2 (C-OH)	C3 (O-C-O/C=O)	C4 (O-C=O)	C5 (CF_2_)	C6 (CF_3_)
Before treatment	32.81	51.89	11.16	4.08	-	-
After treatment	3.24	8.24	7.08	4.14	75.64	1.66

## Data Availability

Not applicable.

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
