# Peer review of "Cellulose from Annual Plants and Its Use for the Production of the Films Hydrophobized with Tetrafluoroethylene Telomers"

_molecules, 2022, doi:10.3390/molecules27186002_

Round 1

Reviewer 1 Report

The article reports on the production of cellulose from hogweed which is an interesting alternative to the use of trees. Though hogweed must be used very carefully and many countries limit its use now due to its high invasive character. 

Cellulose extracted from hogweed is purified and bleached without using chlorine-containing reactants reaching a very high whiteness index. 

Though interesting and of wide application, the work should be better described, both in terms of used methods and found results.

Concerning the methods, for example, the deconvolution procedure for XPS spectra is not described (gaussian curves?), as well as the details for IR measurements (resolution, number of scans, etc.). How was the degree of crystallinity determined from DSC measurements? And the bound water? Even though the used reference is cited, the procedure should be described. 

Concerning the results, the overall DSC curve should be shown. I guess that the crystallinity degree is determined by the melting peak area.

Page 7, line 221: the description of glass transition from a solid glassy state to an elastic one is not 100% correct. Elasticity is the capability of a solid material to be deformed and then recover the original shape after the stress is removed. In this sense, steel is more elastic than rubber. Above the glass transition temperature, the amorphous part of materials acquires a viscous or rubbery state. Rubbery is not a synonym for elastic.

Be careful with grammar and typos, some sentences are hardly understandable, like the sentence between lines 301 and 302. On page 4, line 134 5C should be changed to 5 °C.

The word "telomere" is almost always used instead of "telomer", but telomeres are segments of DNA which, I don't think, are used here. English must therefore be carefully reviewed.

Author Response

Reply to Reviewer #1 Comments

Reviewer #1:

  1. The article reports on the production of cellulose from hogweed which is an interesting alternative to the use of trees. Though hogweed must be used very carefully and many countries limit its use now due to its high invasive character.

Authors reply:

We agree with the comment of the Reviewer.

Reviewer #1:

  1. Cellulose extracted from hogweed is purified and bleached without using chlorine-containing reactants reaching a very high whiteness index.

Authors reply:

We agree with the comment of the Reviewer.

Reviewer #1:
3. Though interesting and of wide application, the work should be better described, both in terms of used methods and found results.

Authors reply:

We agree with the comment of the Reviewer.

Reviewer #1:
4. Concerning the methods, for example, the deconvolution procedure for XPS spectra is not described (gaussian curves?), as well as the details for IR measurements (resolution, number of scans, etc.). How was the degree of crystallinity determined from DSC measurements? And the bound water? Even though the used reference is cited, the procedure should be described. 

Authors reply:

Thanks for this comment. In the new version of the manuscript, the deconvolution procedure for XPS spectra and details for IR measurements are described in the experimental part. We have not determined the degree of crystallinity (DC) from DSC measurements. The DC value (70%) given in Section 3.5 (line 244), which was used to select the dependence Tg = Tg(CH2O), was calculated by averaging the values obtained from IR (75%) and XRD (65%). The corresponding addition is made in the new version of the manuscript.

Reviewer #1:
5. Concerning the results, the overall DSC curve should be shown. I guess that the crystallinity degree is determined by the melting peak area.

Authors reply:

In fig. 4 shows overall DSC curves for studied HogC samples in the temperature range from –10 to + 200 °С. Heating to temperatures above 200 °C leads to thermal degradation of the cellulose. We used DSC measurements only to determine glass transition temperatures but not crystallinity degree for HogC samples.

Reviewer #1:
6. Page 7, line 221: the description of glass transition from a solid glassy state to an elastic one is not 100% correct. Elasticity is the capability of a solid material to be deformed and then recover the original shape after the stress is removed. In this sense, steel is more elastic than rubber. Above the glass transition temperature, the amorphous part of materials acquires a viscous or rubbery state. Rubbery is not a synonym for elastic.

Authors reply:

We agree with the comment of the Reviewer.
“the transition of a polymer from a solid glassy state to an elastic one” replaced by  “the transition of a polymer from a solid glassy state to a rubber one”.
See updated text.

Reviewer #1:
7. Be careful with grammar and typos, some sentences are hardly understandable, like the sentence between lines 301 and 302. On page 4, line 134 5C should be changed to 5 °C.

Authors reply:

Thanks for this comment. In the new version of the manuscript: a) the sentence between lines 301 and 302 rewritten as follows «Note also that in ideal cellulose there are no carbon atoms, which have only carbon or hydrogen atoms in the nearest environment (see the chemical structure of the polysaccharide and the assignment of peaks in Table 7)»; b) 5C is replaced by 5 °C (on page 4, line 134).

Reviewer #1:
8. The word "telomere" is almost always used instead of "telomer", but telomeres are segments of DNA which, I don't think, are used here. English must therefore be carefully reviewed.

Authors reply:

1 - A telomere is a region of repetitive DNA sequences at the end of a chromosome. The plural of telomere is telomeres

2 – But in radiation chemistry: the process of radiation telomerization, the chain carrier is telogen, the reaction product is telomer [T1-T3]. The plural of telomer is telomers

T1. David С., Gosselain H.A. Etude de la reaction de telomerisation de l`ethylene et du tetrachlorure initiee par rayonnement gamma. Tetrahedron. 1962, v.18, p. 639.

T2. D. P. Kiryukhin, G. A. Kichigina, and P. P. Kushch. Fluoropolymer Composite Materials and Protective Coatings for the Extreme Conditions of the Arctic Zone. Russian Journal of General Chemistry, 2021, Vol. 91, No. 12, pp. 1–9.

T3. S. Barisci, R. Suri “Removal of polyfluorinated telomer alcohol by Advanced Oxidation Processes (AOPs) in different water matrices and evaluation of degradation mechanisms”, Journal of Water Process Engineering 39 (2021) 101745

Thanks to the Reviewer for helpful comments!

Reviewer 2 Report

The manuscript entitled “Cellulose from annual plants and its use for the production of the films hydrophobized with tetrafluoroethylene telomers” is good example of work about cellulose isolation from Sosnovsky hogweed. Comments regarding this manuscript are:

1.      It is not clear for what application author is looking for with this HogC cellulose? It should be specified, and some experiments should be included in this manuscript and should compare the results with commercial/standard celluloses.

2.      Again, why hydrophobization? For what application? Why telomere of tetrafluoroethylene rather than other hydrophobic agents? There are many biobased hydrophobic agents available also.

3.      In line no. 39-40, author mentioned “it is believed that instead of trees ………. annual plants can be used”. Author needs to justify it. None of these references makes any sense. How small annual plant can supply to the huge market demand of cellulose. Some figures or statistics should be added. Some comparative information about how much cellulose can be obtained from annual plant compared to trees? This information is very crucial to justify the goal of this research. The introduction needs to improve.

4.      The degree of crystallinity is not absolute value. It is more a relative value. Thus, a reference sample should be there in order to understand how the process is affecting the cellulose structure.

5.      To analyze bound water, author can try KF titration.

6.      Author should analyze lignin, cellulose, and other sugar content of the weed sample before and after the treatment.

7.      Some electron microscopy studies of cellulose sample should be included to understand the detailed structure of the cellulose

8.      Degree of polymerization of the cellulose sample should be reported as it is one of important parameter to analyze quality of the cellulose fibers.

Author Response

Reply to Reviewer #2 Comments

Reviewer #2:

  1. It is not clear for what application author is looking for with this HogC cellulose? It should be specified, and some experiments should be included in this manuscript and should compare the results with commercial/standard celluloses.

Authors reply:

A possible field of application for HogC is traditional for cellulose. These are artificial fibers and paper. This conclusion can be drawn on the basis of the study of isolated cellulose by NMR spectroscopy, which showed that the composition of HogC cellulose is close to commercial/standard celluloses (see new version of the manuscript).

Reviewer #2:

  1. Again, why hydrophobization? For what application? Why telomere of tetrafluoroethylene rather than other hydrophobic agents? There are many biobased hydrophobic agents available also.

Authors reply:

Hydrophobization is an important way to control the properties of cellulosic materials. In the manuscript, this circumstance was noted by the phrase that "water sensitivity limits the durability of cellulose products [15]" (lines 66-67). We agree with the Reviewer's remark that there are a lot of water repellents. This is the first stage of work for us and in the future we plan to test other water repellents. In the present work, we used new tetrafluoroethylene telomers synthesized using the radiation-chemical method of initiating the reaction (gamma radiation). The developed method makes it possible to synthesize solutions of TFE telomers with different lengths of the tetrafluoroethylene block (n) and different terminal functional units (R1, R2) depending on the solvent used: R1(C2F4)n R2. The developed technology for the production of TFE telomeric solutions using radiation-chemical initiation and the creation of protective fluoropolymer coatings on their basis is aimed at a wide area of ​​commercial use. Solutions of TFE telomers can be used to impregnate fabrics, wood, asbestos, cement, create coatings on metal and ceramic products and other objects, to give them chemical and corrosion resistance, water-repellent, anti-friction, anti-wear properties. The use of TFE telomer solutions to create protective coatings makes it possible to overcome the technological and operational limitations experienced by traditional technologies for applying fluoropolymer coatings (suspensions, plasma-chemical methods, condensation of pyrolysis products, the use of powders, etc.). The proposed technologies do not yet have analogues in the world practice and are protected by a number of patents of the Russian Federation.

Reviewer #2:
3. In line no. 39-40, author mentioned “it is believed that instead of trees ………. annual plants can be used”. Author needs to justify it. None of these references makes any sense. How small annual plant can supply to the huge market demand of cellulose. Some figures or statistics should be added. Some comparative information about how much cellulose can be obtained from annual plant compared to trees? This information is very crucial to justify the goal of this research. The introduction needs to improve. 

Authors reply:

In the area of the Caucasus foothills the plant reaches approximately 1-1.5 m in height, whereas in Poland its size is significantly larger, up to 3-3.5 m. It has thickly ridged, hollow stem, up to 12 cm in diameter. The leaves are palmate and reach up to 2 m in length (O. Jakubowicz et al - Annals of Agricultural and Environmental Medicine 2012, Vol 19, No 2, 327-328). The biomass can reach 2.3–4.6 kg/m2. (L.M. Abramova et al, Russ J Biol Invasions 2021, 12, 127–135). Even with a minimum yield of 2 kg/m2, we will get 20 tons of biomass from one ha of hogweed. The annual growth of wood in our latitudes is about 4 m3/ha, which gives about 6 tons of biomass (this is an average value). Therefore, the production of pulp from hogweed (even at 35% yield) appears to be more efficient in terms of land use compared to production from wood.

We also note here that, according to some estimates that can be found on the Internet, hogweed occupies more than a million hectares in the European part of Russia.

Corresponding changes have been made to the Introduction in the new version of the manuscript.

Reviewer #2:
4. The degree of crystallinity is not absolute value. It is more a relative value. Thus, a reference sample should be there in order to understand how the process is affecting the cellulose structure. 

Authors reply:

We agree with the reviewer's remark. In the new version of the manuscript, the degree of crystallinity of HogC (48%), which was determined from the IR spectrum, is compared with that of cotton (65%). We note here for the reviewer that the cotton fiber for IR was taken directly from an open cotton bowl.

Reviewer #2:
5. To analyze bound water, author can try KF titration.

Authors reply:

Karl Fischer titration we did not. But we have TGA, DSC and IR data. We thought this was enough.

Reviewer #2:
6. Author should analyze lignin, cellulose, and other sugar content of the weed sample before and after the treatment.

Authors reply:

In order to answer this remark of the reviewer, we conducted a study of the HogC sample by solid-state NMR spectroscopy (see the corresponding inserts in the experimental part and in the results). It was found that the composition of the test sample corresponds with good accuracy to the chemical formula of cellulose. There were no peaks that could be attributed to lignin in the NMR spectrum of the studied HogC sample. In view of the foregoing and the output indicated in the experimental part, it can be argued that the cellulose content in hogweed is at least 35 wt. %.  

Reviewer #2:
7. Some electron microscopy studies of cellulose sample should be included to understand the detailed structure of the cellulose.

Authors reply:

We have photographs of our samples obtained using optical microscopy.But we do not present them in this publication.But we present one of these photographs for the reviewer here in the Authors reply.We don't have TEM photos yet.You can't do everything in one job.

Optical photographs of cellulose isolated from a hogweed stem. Cellulose was stained with methylene blue for clarity.

Reviewer #2:
8. Degree of polymerization of the cellulose sample should be reported as it is one of important parameter to analyze quality of the cellulose fibers.

Authors reply:

The authors of this manuscript, unfortunately, do not have in their arsenal a technique for determining the degree of polymerization (DP) of the cellulose samples. We gave our samples for DP determination to other organizations. The data obtained differed by more than 2 times (from 1590 to 4400), despite the fact that the same technique (viscometry technique) was used. We are not yet ready to publish the results. Possibly, such a scatter is due to the fact that in our sample, in addition to the cellulose of the fibrous part, there is also cellulose of the parenchyma. In future work, we plan to separate these contributions.  

Thanks to the Reviewer for helpful comments!

Reviewer 3 Report

1)     Cellulose HogC. Please first write the full name, then the abbreviation, after that the abbreviation can be used in the rest of the parts

2)     The details of each chemical should be mentioned ( Company, city, country)

3)     Please mention the specification of water that was utilized.

4)      More details are required here"  The system was stirred at room temperature and irradiated with 60Co γ-rays at the Gammatok-100 unique research facility at a 112 dose rate of 3.2 Gy/s.

5)     Usually, Cr is a symbol of Chromium therefore I suggest changing the sample that you use for example cotton (Cr-I),

6)     In figure 4 the numbers 1, 2, and 3. Please put a description of these numbers and what they indicate clearly

7)     3.6. Please mention the full name folwed by the abbreviation (CWA)

8)     3.7. Please mention the full name folwed by the abbreviation (XPS)

9)     I propose to write the various chemical reactions that occur between cellulose and the materials used and discuss the mechanical effects on the properties of the resulting materials.

10)  Please discuss the environmental impact of this process. Is it more appropriate than the well-established one? 

Author Response

Reply to Reviewer #3 Comments

Reviewer #3:

  1. Cellulose HogC. Please first write the full name, then the abbreviation, after that the abbreviation can be used in the rest of the parts.

Authors reply:

Thanks for this comment. In the new version of the manuscript, the abbreviation is given after the full name.

Reviewer #3:

  1. The details of each chemical should be mentioned ( Company, city, country).

Authors reply:

Nitric acid (concentration 70% GOST 11125-84) was obtained from JSC Base No. 1 of Chemical Reactives, Staraya Kupavna, Russia. Hydrogen peroxide (concentration 33%, TU 20.13.63-207-44493179-2016) and potassium hydroxide (chemically pure) were provided by OOO EKOS-1, Russia, Staraya Kupavna. Tetrafluoroethylene (С2F4, TFE), Flutec PP3 (perfluoro-1,3-dimethylcyclohexane, C8F16) and trifluorotrichloroethane (C2F3Cl3, Freon 113), produced by the Polymer Plant of the Kirovo-Chepetsk Chemical Integrated Works, acetone (C3H6O) (Aldrich, Germany) and  gaseous ammonia NH3 (United Chemical Company Shchekinoazot, Tula, Russia) were used for the radiation synthesis of telomers. The reagents were used without further purification. Double-distilled water was used in the work. The reagents were used without further purification. We have made appropriate changes in the new version of the manuscript.

Reviewer #3:
3. Please mention the specification of water that was utilized.

Authors reply:

Double-distilled water was used in the work.

Reviewer #3:
4. More details are required here"  The system was stirred at room temperature and irradiated with 60Co γ-rays at the Gammatok-100 unique research facility at a 112 dose rate of 3.2 Gy/s.

Authors reply:

Here, the reviewer's remark included the line number (112) where this sentence was written. Perhaps this confused the reviewer. Once again, especially for the reviewer, we note that the installation is described in detail in reference 49 in the old version of the manuscript.

The main lines of research using the Gammatok100 concern the development of new basic and applied areas of radiation chemistry: radiation chemistry of polymer–monomer systems; radiation cryochemistry; radiation chemistry of nanomaterials and natural materials; radiation resistance of materials’ radiation-chemical technology of new processes; an development of experimental and theoretical foundations of fabrication of composite materials with improved mechanical characteristics, including using the combined effects of high-energy radiation (gamma-radiation, accelerated particles, laser radiation). A number of important fundamental results have been obtained.

Reviewer #3:
5. Usually, Cr is a symbol of Chromium therefore I suggest changing the sample that you use for example cotton (Cr-I),

Authors reply:

We agree with the comment of the Reviewer. In the new version of the manuscript, we have used the notation suggested by the reviewer.

Reviewer #3:
6. In figure 4 the numbers 1, 2, and 3. Please put a description of these numbers and what they indicate clearly

Authors reply:

In the new version of the manuscript, we have given a description of these numbers in the figure's caption.

Reviewer #3:
7. 3.6. Please mention the full name folwed by the abbreviation (CWA)

Authors reply:

Thanks for this comment. In the new version of the manuscript, the abbreviation is given after the full name.

Reviewer #3:
8. 3.7. Please mention the full name folwed by the abbreviation (XPS)

Authors reply:

Thanks for this comment. In the new version of the manuscript, the abbreviation is given after the full name.

Reviewer #3:
9. I propose to write the various chemical reactions that occur between cellulose and the materials used and discuss the mechanical effects on the properties of the resulting materials.

Authors reply:

Traditional methods of pulping are carried out under rather severe conditions: at temperatures up to 180 °C and high pressures. The main methods for producing cellulose are the sulfate method (treatment with aqueous solutions of NaOH and Na2S) and the sulfite method (treatment with aqueous solutions of calcium, magnesium, sodium hydrosulfite containing free SO2), which are used in the production of the most common wood pulp.

In this work, the treatment of the starting material (stems of Sosnovsky hogweed) with weak nitric acid results in nitration and oxidation of lignin, accompanied by its destruction and dissolution at the stage of alkaline extraction. The main technological advantages of nitric acid cooking are: speed, moderate temperature and the absence of high pressure. Low cooking temperature and alkaline extraction (80-100°C) helps to reduce energy consumption. Carrying out cooking at atmospheric pressure allows you to simplify the cooking equipment.

Bleaching is an important stage in pulp refining. The most common reagents used in the bleaching process at most pulp and paper mills are chlorine, hypochlorous acid salts - hypochlorites, and chlorine dioxide. Bleaching is carried out by a multi-stage (combined) method.

The pulp is alternately exposed to various bleaching agents with intermediate washings between the individual stages. The classic bleaching scheme for wood (pine) pulp includes: chlorination - alkaline treatment - alkaline refining - hypochlorite bleaching - bleaching with chlorine dioxide - acidification. The use of chlorine is not environmentally safe, and alternative methods for bleaching pulp with hydrogen peroxide, ozone, active oxygen, etc. are currently being actively developed.

When bleaching with hydrogen peroxide, the direct bleaching effect on cellulose, or rather, on its residual lignin, is exerted by the peroxide ion resulting from the hydrolytic dissociation of peroxide: Н2О2 ↔ Н+ + НО2.   The dissociation of peroxide is facilitated by an alkaline environment, since hydrogen ions are neutralized in this case, and the equilibrium of the hydrolysis reaction shifts to the right. Therefore, bleaching with peroxide is carried out in an alkaline environment, at pH = 10-11. The use of hydrogen peroxide as a bleaching agent made it possible to obtain bleached pulp from Sosnowski's hogweed with a high (over 92%) level of whiteness. At the same time, as an advantage over the classical scheme of chlorine bleaching of pulp, in addition to higher environmental friendliness, one can indicate the simplicity of the method, because the process is performed in just one-step.

Reviewer #3:
10. Please discuss the environmental impact of this process. Is it more appropriate than the well-established one? 

Authors reply:

When bleaching with hydrogen peroxide, the direct bleaching effect on cellulose, or rather, on its residual lignin, is exerted by the peroxide ion resulting from the hydrolytic dissociation of peroxide: Н2О2 ↔ Н+ + НО2.   The dissociation of peroxide is facilitated by an alkaline environment, since hydrogen ions are neutralized in this case, and the equilibrium of the hydrolysis reaction shifts to the right. Therefore, bleaching with peroxide is carried out in an alkaline environment, at pH = 10-11. The use of hydrogen peroxide as a bleaching agent made it possible to obtain bleached pulp from Sosnowski's hogweed with a high (over 92%) level of whiteness. At the same time, as an advantage over the classical scheme of chlorine bleaching of pulp, in addition to higher environmental friendliness, one can indicate the simplicity of the method, because the process is performed in just one-step.

See also the answer to remark 9.

Thanks to the Reviewer for helpful comments!

Round 2

Reviewer 1 Report

The authors answered all questions and performed all corrections required. The missing information was also added. The addition of NMR improves the results section by completing the characterization framework and contributes significantly to support the conclusions.

Author Response

Reply to Reviewer #1 Comments and Suggestions

Reviewer #1:

The authors answered all questions and performed all corrections required. The missing information was also added. The addition of NMR improves the results section by completing the characterization framework and contributes significantly to support the conclusions.

Authors reply:

We agree with the comment of the Reviewer.

 Thanks to the Reviewer for helpful comments!

Reviewer 2 Report

The author has addressed some comments that was raised by reviewer. However, the manuscript still requires some additions. The comments regarding this manuscript are:

1.      Author has mentioned that the HogC application is traditional cellulose. And author included some solid-state NMR experiments to compare the composition with commercial cellulose. Unfortunately, solid-state NMR gives no information about the quality of cellulose as well as signal of solid-state NMR are very weak. Nevertheless, the crystallinity index measurement by solid-state NMR is not reliable at all compared to XRD. I am completely disagreed with author’s decision to compare cellulose quality by just solid-state NMR. To compare cellulose pulp quality with commercial sample, author must do some standard analysis those are used for many decades in the field of pulp and paper industries. These tests are very standard (ISO, TAPPI etc) and methods are available online. Author must include at least some tests or experiments related to fiber dimensions or quality or similar type analysis.

2.      It was already pointed out that author need to analyze lignin, cellulose or hemicellulose or other sugar content before and after the treatment. This is not for the purity assignment; this is mainly how the process is affecting the whole biomass. Solid state NMR analysis does not make any senses.

3.      Electron microscopy should be included in this manuscript to characterize the isolated cellulose. Incomplete characterization of cellulose will not create enough novel impact to publish a paper in renowned journal like “Molecules”.

4.      Author measured degree of polymerization via viscometric method (not mentioned which method) however the results were not reproducible this is probably due to the viscosity of the pulp values are not reproducible. DP (degree of polymerization) totally depends on the values of viscosity (if measured via viscometric method). The parameters that can affects viscosity is sugar residue, fiber size, protein, or other nitrogenous compounds etc. Thus, the cellulose needs to fully characterize before measuring viscosity/DP. Author should use the standard method (e.g., TAPPI etc) which available online. However, there are some reproducibility and repeatability range for this analysis because the method identifies the viscosity of 0.5% pulp solution in cupriethylenediamine (CED) reagent.

The manuscript needs these additions to create enough novel impact to publish in high quality journal like “Molecules”.

Author Response

Reply to Reviewer #2 Comments

Reviewer #2:

Author has mentioned that the HogC application is traditional cellulose. And author included some solid-state NMR experiments to compare the composition with commercial cellulose. Unfortunately, solid-state NMR gives no information about the quality of cellulose as well as signal of solid-state NMR are very weak. Nevertheless, the crystallinity index measurement by solid-state NMR is not reliable at all compared to XRD. I am completely disagreed with author’s decision to compare cellulose quality by just solid-state NMR. To compare cellulose pulp quality with commercial sample, author must do some standard analysis those are used for many decades in the field of pulp and paper industries. These tests are very standard (ISO, TAPPI etc) and methods are available online. Author must include at least some tests or experiments related to fiber dimensions or quality or similar type analysis.

Authors reply:

We do not agree with the Reviewer in his assessment of the possibilities of the NMR method. At the same time, we are not opposed to traditional methods, "used for many decades in the cellulose and paper industry." Taking into account the recommendation of the Reviewer, we analyzed the content of lignin by the modified Klason method described in RF patent No. 2405877 (RU 2405877 METHOD OF DETECTING LIGNIN CELLULOSE INTERMEDIATE PRODUCTS) both in the original sample of hogweed and bleached cellulose. The method is based on the dissolution of the carbohydrate part of the lignin-containing raw material in 72% sulfuric acid, followed by nitration with concentrated nitric acid, alkalization, and spectrophotometric determination of the amount of lignin. The content of lignin in the initial material of hogweed according to the results of measurements was 31.8 wt %, the content of lignin in bleached cellulose could not be determined, because the peak characteristic of lignin in the region of 315–340 nm cannot be separated from the sloping background of water. Our noise level estimate indicates that the lignin content of the studied cellulose is less than 0.5 wt. %. The high level of whiteness of the obtained sample indicates its almost complete absence in the resulting cellulose.

Reviewer #2:

  1. It was already pointed out that author need to analyze lignin, cellulose or hemicellulose or other sugar content before and after the treatment. This is not for the purity assignment; this is mainly how the process is affecting the whole biomass. Solid state NMR analysis does not make any senses.

Authors reply:

See answer to previous comment.

Reviewer #2:
Electron microscopy should be included in this manuscript to characterize the isolated cellulose. Incomplete characterization of cellulose will not create enough novel impact to publish a paper in renowned journal like “Molecules”.

Authors reply:

Electron microscopy is a powerful research tool. It includes several methods. The Reviewer's recommendation does not indicate the problem that we must solve with electron microscopy. In the previous answer, we provided photographs of cellulose isolated by us, which were obtained using optical microscopy. From it was seen that the resulting materials have a high aspect ratio. It is clear that at high magnification we will not be able to see the entire length of the fibers. From optical microscopy, we also see that the thickness of the fibers varies over a fairly wide range. In principle, these limits can be refined using electron microscopy. Using acid hydrolysis, cellulose nanocrystals can be isolated and studied. But this is a separate big task. We took this Reviewer's remark seriously. We currently have a large array of SEM photographs of both white pulp and carbonized pulp. It is not possible to analyze the available photographs at a good level in the short period of time that we were given in response. Moreover, we are not ready to change our manuscript accordingly. Therefore, especially for the Reviewer, we present here 2 photographs. 1 photo of white pulp and 1 photo of carbonized pulp. We believe that we will use these photographs in one of the following works.

SEM Photo of white pulp

SEM Photograph of carbonized (black) pulp

Reviewer #2:
4. Author measured degree of polymerization via viscometric method (not mentioned which method) however the results were not reproducible this is probably due to the viscosity of the pulp values are not reproducible. DP (degree of polymerization) totally depends on the values of viscosity (if measured via viscometric method). The parameters that can affects viscosity is sugar residue, fiber size, protein, or other nitrogenous compounds etc. Thus, the cellulose needs to fully characterize before measuring viscosity/DP. Author should use the standard method (e.g., TAPPI etc) which available online. However, there are some reproducibility and repeatability range for this analysis because the method identifies the viscosity of 0.5% pulp solution in cupriethylenediamine (CED) reagent.

Authors reply:

We agree with the reviewer's remark.

The main data on the degree of polymerization were obtained by the method described in the work. We note here that for samples with a low degree of polymerization, peaks appear in the NMR spectrum corresponding to the end groups. The absence of such in the spectra obtained by us means that the degree of polymerization of our samples is quite high.

Thanks to the Reviewer for helpful comments!

Reviewer 3 Report

Thanks. Now the paper is better than before.

Author Response

Reply to Reviewer #3 Comments

Authors reply:

Thanks to the Reviewer for helpful comments!